# A concerted increase in readthrough and intron retention drives transposon expression during aging and senescence

Kamil Pabis[1,2,3]*, Diogo Barardo[2,3], Olga Sirbu[4], Kumar Selvarajoo[4,5,6,7], Jan Gruber[1,8], Brian K Kennedy[1,2,3]

[1]Department of Biochemistry, Yong Loo Lin School of Medicine, National University of Singapore, Singapore, Singapore; [2]Healthy Longevity Translational Research Programme, Yong Loo Lin School of Medicine, National University of Singapore, Singapore, Singapore; [3]Centre for Healthy Longevity, National University Health System, Singapore, Singapore; [4]Bioinformatics Institute, Agency for Science, Technology and Research (A*STAR), Singapore, Singapore; [5]Singapore Institute of Food and Biotechnology Innovation, Agency for Science, Technology and Research (A*STAR), Singapore, Singapore; [6]NUS Synthetic Biology for Clinical and Technological Innovation (SynCTI), National University of Singapore, Singapore, Singapore; [7]School of Biological Sciences, Nanyang Technological University, Singapore, Singapore; [8]Science Divisions, Yale-NUS College, Singapore, Singapore

*For correspondence:
Kamil.pabis@gmail.com

Competing interest: The authors declare that no competing interests exist.

**Abstract** Aging and senescence are characterized by pervasive transcriptional dysfunction, including increased expression of transposons and introns. Our aim was to elucidate mechanisms behind this increased expression. Most transposons are found within genes and introns, with a large minority being close to genes. This raises the possibility that transcriptional readthrough and intron retention are responsible for age-related changes in transposon expression rather than expression of autonomous transposons. To test this, we compiled public RNA-seq datasets from aged human fibroblasts, replicative and drug-induced senescence in human cells, and RNA-seq from aging mice and senescent mouse cells. Indeed, our reanalysis revealed a correlation between transposons expression, intron retention, and transcriptional readthrough across samples and within samples. Both intron retention and readthrough increased with aging or cellular senescence and these transcriptional defects were more pronounced in human samples as compared to those of mice. In support of a causal connection between readthrough and transposon expression, analysis of models showing induced transcriptional readthrough confirmed that they also show elevated transposon expression. Taken together, our data suggest that elevated transposon reads during aging seen in various RNA-seq dataset are concomitant with multiple transcriptional defects. Intron retention and transcriptional readthrough are the most likely explanation for the expression of transposable elements that lack a functional promoter.

## eLife assessment

This study presents **fundamental** findings on the role of transcription readout and intron retention in transposon expression during aging in mammals. The evidence supporting the claims of the authors is **compelling**, strongly supporting the authors' claims. The work will be of interest to scientists studying aging, transcription regulation, and epigenetics.

## Introduction

Repetitive DNA makes up at least 50% of the human genome (*de Koning et al., 2011*) and has been linked to genomic instability, cancer and – somewhat less consistently – species longevity (*Pabis, 2021*; *Khristich and Mirkin, 2020*). Transposons are one particularly abundant class of repetitive sequences found in the nuclear genome of eukaryotes and are strongly associated with cancer-causing structural variants like DNA deletions, insertions, or inversions (*Rodriguez-Martin et al., 2020*). Although transposons are often considered parasitic, harmful or, at best, neutral recent evidence suggests they may contribute to genomic diversification on evolutionary timescales and to the cellular stress-response by providing transcription factor-binding sites (*Villanueva-Cañas et al., 2019*).

The major transposon families in the human genome are long interspersed nuclear element (LINE), short interspersed nuclear element (SINE), long terminal repeat (LTR) transposons, and DNA transposons comprising 21%, 11%, 8%, and 3% of the human genome, respectively (*Kazazian and Moran, 2017*).

In recent years, transposon expression, particularly of elements belonging to the LINE-1 family, has been hypothesized to be a near universal marker of aging. Several lines of evidence support a link between transposon reactivation and aging (*Gorbunova et al., 2021*).

Using RNA-seq, for example, to quantify reads mapping to transposable elements it was shown that expression of these was higher in 10- vs 1-day-old nematodes (*LaRocca et al., 2020*), 40- vs 10-day-old flies (*Wood et al., 2016*) and in muscle and liver of aged mice (*De Cecco et al., 2019*). Transposon expression also increases during in vitro senescence (*Colombo et al., 2018*) and in fibroblasts isolated from people between the ages of 1 and 96 years (*LaRocca et al., 2020*), whereas lifespan extending mutations and interventions in mice reduce transposon expression (*Wahl et al., 2021*).

However, not all studies report such consistent age-related increases across all transposon classes (*Ghanam et al., 2019*). Moreover, it would be a mistake to conflate changes in RNA-seq-, RNA-, DNA-, and protein-based measurements of transposon expression as evidence for one and the same phenomenon.

Quantification of transposon expression using RNA-seq or PCR techniques remains challenging given their repetitive nature and the fact that 98–99% of all transposons are co-expressed with neighboring transcriptional units (*Deininger et al., 2017*; *Stow et al., 2021*). In fact, only a small fraction of LINE-1 and SINE elements, the latter relying on co-transposition by LINEs, are expressed from a functional promoter. It is believed that such 'hot' LINE-1 loci drive most transposition events and genomic instability (*Deininger et al., 2017*; *Rodriguez-Martin et al., 2020*).

The large number of co-expressed transposons would likely mask any signal from autonomous elements in standard RNA-seq experiments. Therefore, alternative explanations are needed for increased transposon expression in such datasets and these could include age-related changes in transposon adjacent genes for example intron retention and transcriptional readthrough.

Most eukaryotic genes contain introns which need to be removed during a complex co-transcriptional process called splicing. Defects in splicing are the cause of many, often neurologic, hereditary conditions (*Scotti and Swanson, 2016*). One such type of splicing defect is intron retention. Although basal levels of intron retention may be benign or even physiologic, high levels are considered harmful (*Zheng et al., 2020*). Aging is accompanied by many changes to splicing patterns (*Wang et al., 2018*), including intron retention, which is increased during cellular senescence (*Yao et al., 2020*) and in aging *Drosophila*, mouse hippocampus and human prefrontal cortex. In addition, levels in human AD brain tissues are elevated even further than in the aged brain (*Adusumalli et al., 2019*).

Transcription is terminated by the cleavage and polyadenylation machinery when the polymerase transcribes through the polyadenylation signal leading to pre-mRNA cleavage, polymerase pausing, conformational change, and eventually polymerase detachment (*Rosa-Mercado et al., 2021*). However, this termination may fail leading to so-called readthrough transcription which increases during stress conditions (*Rosa-Mercado and Steitz, 2022*) and during cellular senescence, at least for a subset of genes (*Muniz et al., 2017*).

Emerging evidence suggests that intron retention, readthrough, and transposon expression are linked (*Rosa-Mercado et al., 2021*; *Hadar et al., 2022*), but this has not been studied in the context of aging. In this manuscript, we show that intron retention and readthrough taken together can explain most of the apparent increase in transposon expression seen in RNA-seq datasets of aging.

**Table 1.** Fraction of genic or intronic transposons and of transposons downstream (ds) or upstream (us) of genes in two different datasets.

| Dataset | Genic | Intronic | ds | us | ds | us | n |
|---|---|---|---|---|---|---|---|
| All transposons | 0.604 | 0.493 | 0.045 | 0.049 | 0.187 | 0.174 | 4,520,928 |
| Fleischer et al. (aging, significant) | 0.857 | 0.564 | 0.082 | 0.025 | 0.105 | 0.037 | 7673 |
| Colombo et al. (senescence, significant) | 0.698 | 0.540 | 0.107 | 0.053 | 0.177 | 0.118 | 8730 |
| | Within genes | | Wthin 5 kb | | Within 100 kb | | |

## Results

### Genomic location of transposons implicates them as a marker of multiple transcriptional defects

During aging and senescence the expression of all four major transposon families (LTR, DNA, LINE, and SINE) increases. However, such a global increase in transposon expression cannot be explained by transposon biology alone since most transposons are ancient, and, having accumulated many mutations, lack a functional promoter (*Deniz et al., 2019*). Instead, we reasoned that transposon expression is associated with the expression of nearby genes.

Consistently, out of 4.5 million annotated transposons 60% were intragenic (most of them intronic), 10% within 5 kb of a gene and the remaining 30% were intergenic, albeit usually still within 100 kb (*Table 1*). For transposons whose expression significantly changed with age or senescence, the number of intronic and downstream-of-gene transposons is even higher.

Based on these findings, we studied intronic transposons as a potential marker for age-related intron retention and downstream-of-gene transposons as a marker of age-related readthrough transcription.

### Elevated transposon expression and intron retention with age

To confirmed that age-related transposon expression is indeed elevated as shown by *LaRocca et al., 2020*, we analyzed the transcriptomic dataset originally generated by *Fleischer et al., 2018*, which included 143 fibroblasts isolated from donors between the ages of 0–96. This dataset will be hereafter referred to as the 'aging dataset'. We find that transposon expression increases slowly and non-linearly during aging, with a sudden and sharp increase around 80 years of age (*Figure 1A*, *Figure 1—figure supplement 1*) and this increase was highly strand specific (*Figure 1—figure supplement 2*).

Next, we found that intron retention is elevated with aging (*Figure 1B*), consistent with the study by *Yao et al., 2020*, and that samples with high transposon expression also showed elevated intron retention (*Figure 1C*).

In our analysis, we corrected for gene co-expression by normalizing the expression levels of introns and transposons to the expression levels of the closest gene. Another complementary approach would be to plot and compare the raw, unadjusted log-fold changes for genes, introns, transposons, and other elements. In this analysis, we found that, on average, age-related upregulation of transposons and introns is more pronounced than upregulation of genes (*Figure 1—figure supplement 3A*), consistent with *Figure 1A, B*.

Next, we reanalyzed a dataset originally published by *Colombo et al., 2018* which will be referred to as the 'senescence dataset'. Here, we tested whether MDAH041 cells induced into senescence with $H_2O_2$, 5-azacytidine, adriamycin, or serial passaging show elevated transposon expression and intron retention. Consistent with the original findings, all these treatments increased the expression of transposons (*Figure 1D*, *Figure 1—figure supplement 3B*). The retention of introns was also increased (*Figure 1E*) and expression levels of introns and transposons correlated with each other (*Figure 1F*).

Having established that transposons reads and retained introns are elevated with aging and senescence, we tested whether this is conserved between mouse and human. Transposon expression (p < 0.0001) was elevated, and intron retention showed a trend toward elevation (p = 0.14), in the liver of 26-month-old mice, suggesting these age-related transcriptional defects might be conserved

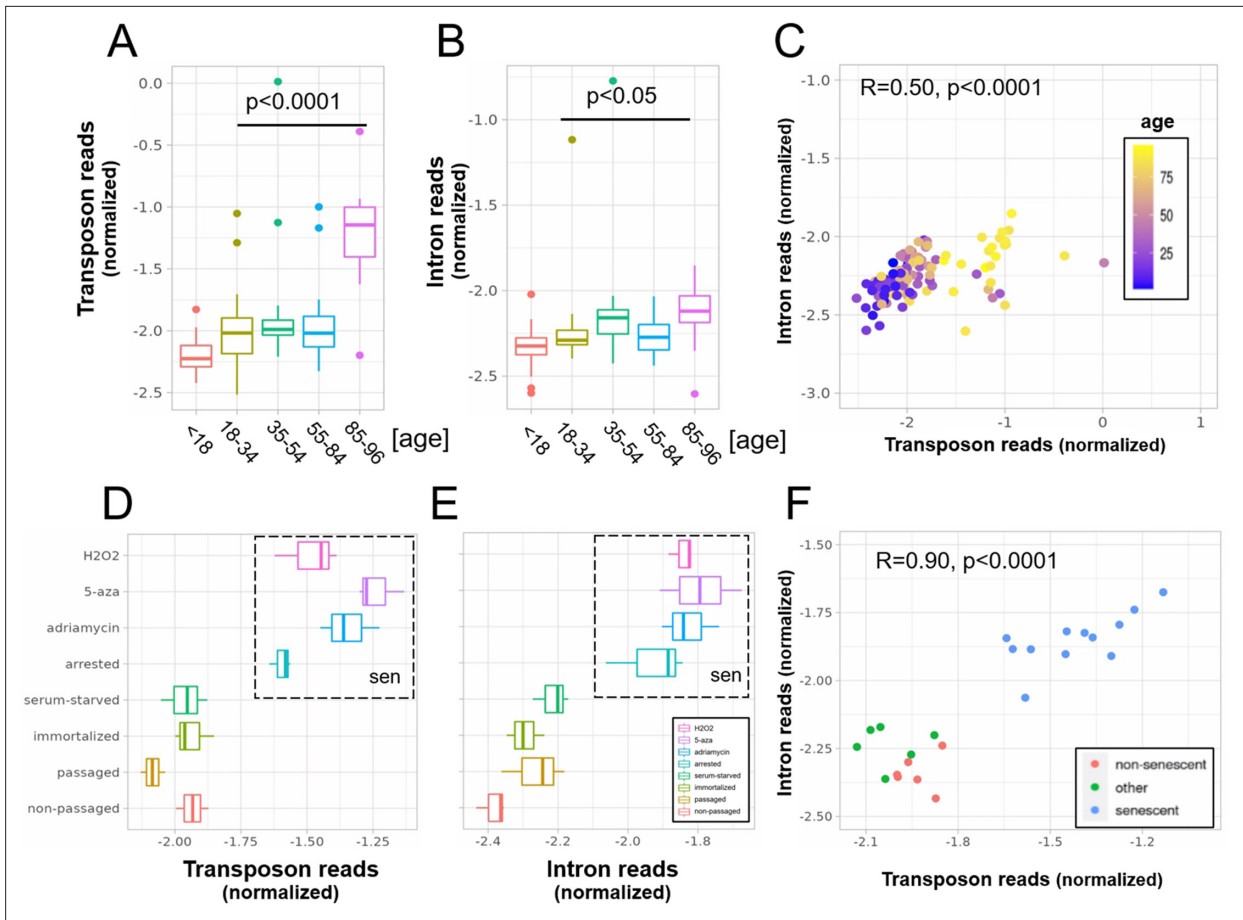

**Figure 1.** A concerted increase in transposon and intron expression with aging and senescence. Transposon (**A**) and intron (**B**) expression is increased in skin fibroblasts isolated from aged individuals. What is more, expression of both transposons and introns shows a significant correlation (**C**). Similarly, transposon (**D**) and intron (**E**) expression is increased in MDAH041 cells induced into senescence (sen) by treatment with drugs or via passaging. Normalized reads from the top 1000 differentially expressed genes, transposons and introns were used in this analysis. (**A**) Transposon reads normalized by the expression of adjacent genes plotted across five age groups (adolescent *n* = 32, young *n* = 31, middle-aged *n* = 22, old *n* = 37, and very old *n* = 21). (**B**) Intron reads normalized by the expression of adjacent genes plotted across five age groups (adolescent *n* = 32, young *n* = 31, middle-aged *n* = 22, old *n* = 37, and very old *n* = 21). (**C**) Scatterplot between transposon and intron expression (normalized as in A and B) for all 143 samples. Each sample is colored by age. (**D**) Transposon reads normalized by the expression of adjacent genes plotted across four senescent conditions ($H_2O_2$, 5-azacytidine, adriamycin, and replicative senescence) and four other conditions (serum-starved, immortalized, intermediate passage, and early passage). *N* = 3 per group. (**E**) Intron reads normalized by the expression of adjacent genes plotted across four senescent conditions ($H_2O_2$, 5-azacytidine, adriamycin, and replicative senescence) and four other conditions (serum-starved, immortalized, intermediate passage, and early passage). *N* = 3 per group. (**F**) Scatterplot between transposon and intron expression (normalized as in E and F) for all 24 samples. Each sample is colored by senescence status.

The online version of this article includes the following figure supplement(s) for figure 1:

**Figure supplement 1.** Aging promotes readthrough, transposon expression, and intron retention.

**Figure supplement 2.** Transposons show strand-specific expression.

**Figure supplement 3.** Stronger upregulation of transposons and introns than of genic transcripts.

**Figure supplement 4.** Increased transposon expression and intron retention in liver of aged mice.

**Figure supplement 5.** Aging promotes readthrough, transposon expression, and intron retention in mouse liver.

**Figure supplement 6.** Correlated age-related increase of transposon and intron loci.

**Figure supplement 7.** Correlated expression of transposon and intron loci.

**Figure supplement 8.** Transposons are depleted at splice junctions.

**Figure supplement 9.** Introns with and without transposons are retained during aging and senescence.

between species (*Figure 1—figure supplement 4A, B*; *Figure 1—figure supplement 5*). In contrast, we found that increased expression of transposons and intron retention in models of cellular senescence is not conserved between species. To the contrary, early passage mouse fibroblasts showed elevated transposon expression and intron retention compared to senescent fibroblasts (*Figure 1—figure supplement 4C, D*), which contrasts with the mouse liver and human senescence data.

## Transposon expression as a marker of intron retention

So far these age-related events have been considered independently, but given the localization of transposons being mostly intragenic we reasoned that intron retention could bias the transposon findings. Indeed, we found that samples with high intron retention also show high transposon expression. This is true both in the aging dataset (*Figure 1C*) and in the cellular senescence dataset (*Figure 1F*), where intronic reads accounted for most of the observed variability in transposon expression (or vice versa). Looking at individual genomic loci, we also found a strong correlation between the log-fold change of a transposon and the log-fold change of the intron it is located in (*Figure 1—figure supplement 6*).

As a more direct test of this hypothesis, we determined the correlation between the counts of retained introns and the counts of intronic transposons for each sample individually. Non-intronic transposons and randomized transposons served as a negative control. Consistent with the log-fold change data, we found a specific correlation between intronic transposon and intron counts in both datasets, with stronger correlations in the senescence dataset (*Figure 1—figure supplement 7*).

Although causality is hard to establish from our datasets, we also tested whether transposon expression could affect intron retention. If transposon expression was causally linked to intron retention, the most likely mechanism would be via an impairment of the splice junction. In this case, we would expect more transposons near the splice junctions of retained (differentially expressed) introns. However, we observed a depletion of transposons at the splice junctions of all introns (*Figure 1—figure supplement 8A*) and of differentially expressed introns (*Figure 1—figure supplement 8B, E*). Transposon densities were also comparable between these two classes of introns (*Figure 1—figure supplement 8C, F*). Moreover, introns with and without expressed transposons showed similar levels of age-related upregulation (*Figure 1—figure supplement 9*). Based on our data, we propose a model (*Figure 2*) where intron retention contributes to age-related upregulation of transposons, rather than transposons contributing to intron retention.

## Expression of extragenic transposons as a marker of readthrough transcription

Most transposons showing increased expression with aging or cellular senescence mapped within genes or introns and their expression changes may be attributed to intron retention. How can we explain expression of the remaining 14–30% of transposons (*Table 1*)?

To answer this question, we analyzed the distribution of transposons mapping to extragenic regions. Out of all the transposons, showing significantly changed expression during aging or cellular senescence, most were located within 5 kb of genes, primarily downstream of genes (*Figure 3A*). The distribution of transposons in the aging and senescence dataset was very similar except for a wider distribution of senescence-associated transposons within a 100-kb window around genes (*Figure 3B*). This contrasts with the almost completely even distribution of randomly permuted transposons.

Given the proximity of significantly changed transposons and genes, this suggested that an age-related increase in transcriptional readthrough could explain elevated transposon expression. To test this, we next quantified readthrough in a 10- to 20-kb region downstream of genes. We chose a region not immediately adjacent to genes to avoid biases through basal levels of readthrough, although the data were comparable for closer distances (*Figure 3—figure supplement 1*).

Indeed, readthrough transcription was increased with aging (*Figure 4A, B*), and in the senescence dataset, levels of readthrough transcription increased with replicative or chemically induced senescence (*Figure 4D, E*). Transposon expression correlated with readthrough levels on a per sample basis in both the aging and the senescence dataset (*Figure 4C, F*).

As shown before for intron retention, these findings may not translate to other species. We found that readthrough was unchanged in the liver of 26-month-old mice (p = 0.22, *Figure 4—figure*

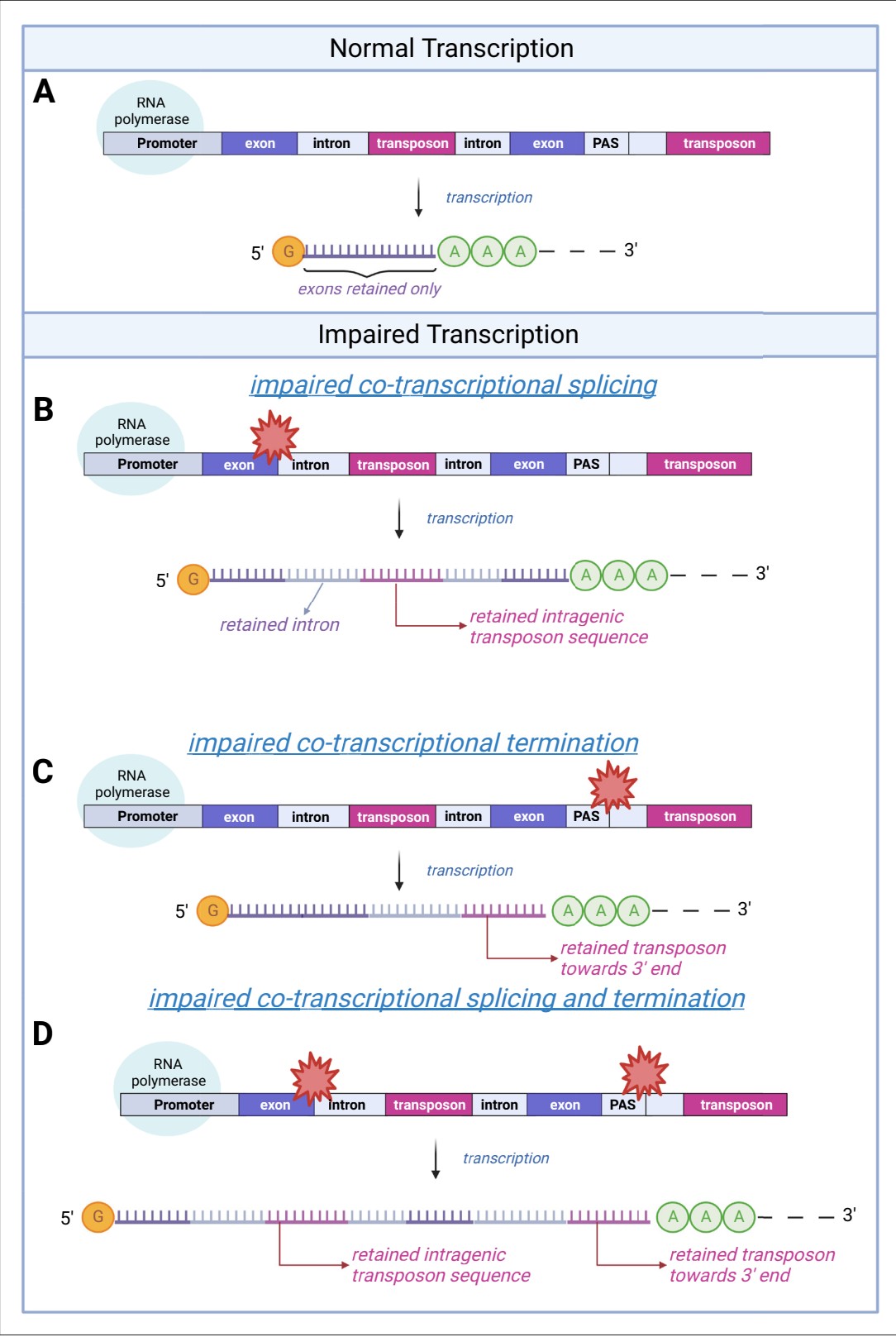

**Figure 2.** A schematic model of transposon expression. In our model, represented in this schematic, transcription (**A**) can give rise to mRNAs and pre-mRNAs that contain retained introns when co-transcriptional splicing is impaired. This is often seen during aging and senescence, and these can contain transposon sequences (**B**). In addition, transcription can give rise to mRNAs and pre-mRNAs that contain transposon sequences toward the 3'-end of

*Figure 2 continued on next page*

the mRNA when co-transcriptional termination at the polyadenylation signal (PAS) is impaired (**C, D**) as seen with aging and senescence. Some of these RNAs may be successfully polyadenylated (as depicted here) whereas others will be subject to nonsense mediated decay. Image created with Biorender.

*supplement 1A*). In addition, readthrough decreased with replicative senescence of mouse fibroblasts, contrary to our expectations (*Figure 4—figure supplement 1C, D*).

Next, we quantified readthrough across a 100-kb region downstream of genes. We found that both aging and senescence lead to readthrough across the whole region, although senescence was associated with more extensive readthrough further downstream of genes (*Figure 4—figure supplement 2*, *Figure 3B*). As another line of evidence we quantified transcription upstream of genes, which is called read-in. This kind of transcription is often attributed to widespread readthrough extending from the end of a gene to the start of another (*Roth et al., 2020*). Consistent with the extensive readthrough levels observed we found elevated read-in transcription during cellular senescence (*Figure 4—figure supplement 3A*) but not aging (*Figure 4—figure supplement 3B*).

Transposon counts followed a similar decaying pattern to readthrough counts downstream of genes (*Figure 4—figure supplement 4*) and genes with high age-related readthrough were associated with strong age-related upregulation of gene proximal transposons (*Figure 4—figure supplement 5*). Intergenic transposons showed such low expression levels that their contribution to total counts was negligible (*Figure 4—figure supplement 6*).

Having shown that samples with higher levels of readthrough also have higher transposon expression (per sample correlation), we asked whether transposons are more often than expected located in readthrough regions (per locus correlation). Indeed as expected, in both datasets, transposons significantly changed with age or senescence were more often found in significant readthrough regions compared to all transposons. Moreover, there was a significant correlation between the log-fold changes of transposons downstream of genes with the log-fold changes of readthrough downstream

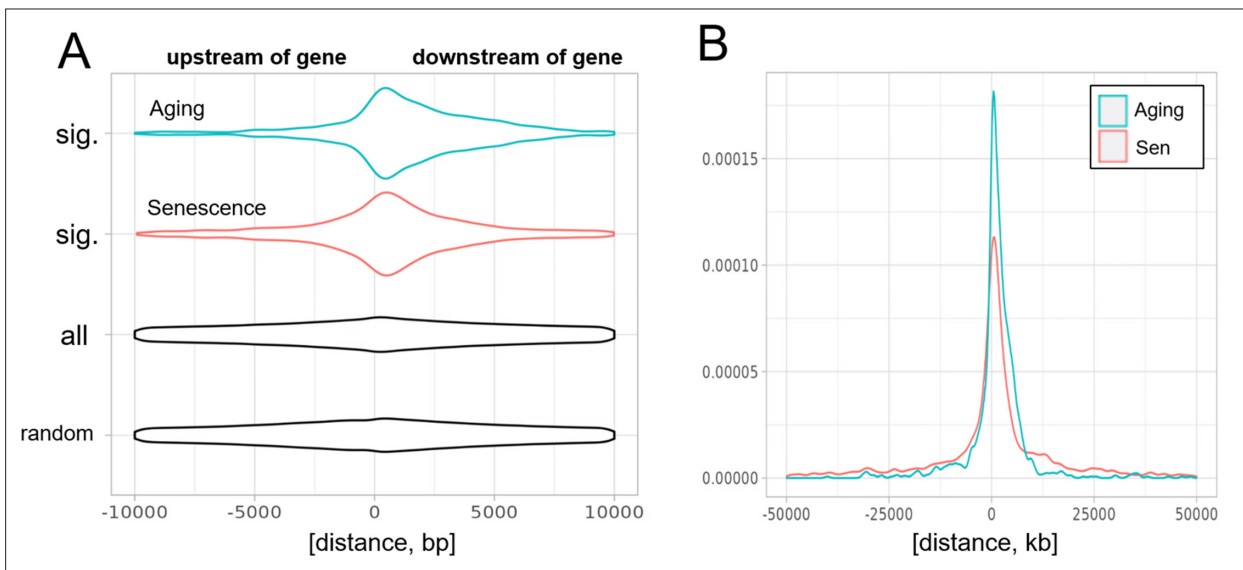

**Figure 3.** Biased distribution of extragenic transposons relative to genes. (**A**) Extragenic transposons with significantly (sig.) changed expression during aging or cellular senescence show a biased distribution, when mapped back onto the genome, with a preference toward a 5-kb region at the 3'-end of genes when compared to all annotated transposons (all) or randomly permuted transposons (random). Permutation was performed using the bedtools shuffle function. (**B**) Extragenic transposons whose expression changes during cellular senescence are spread out further from genes compared to aging-associated transposons.

The online version of this article includes the following figure supplement(s) for figure 3:

**Figure supplement 1.** Increased readthrough levels with aging and senescence.

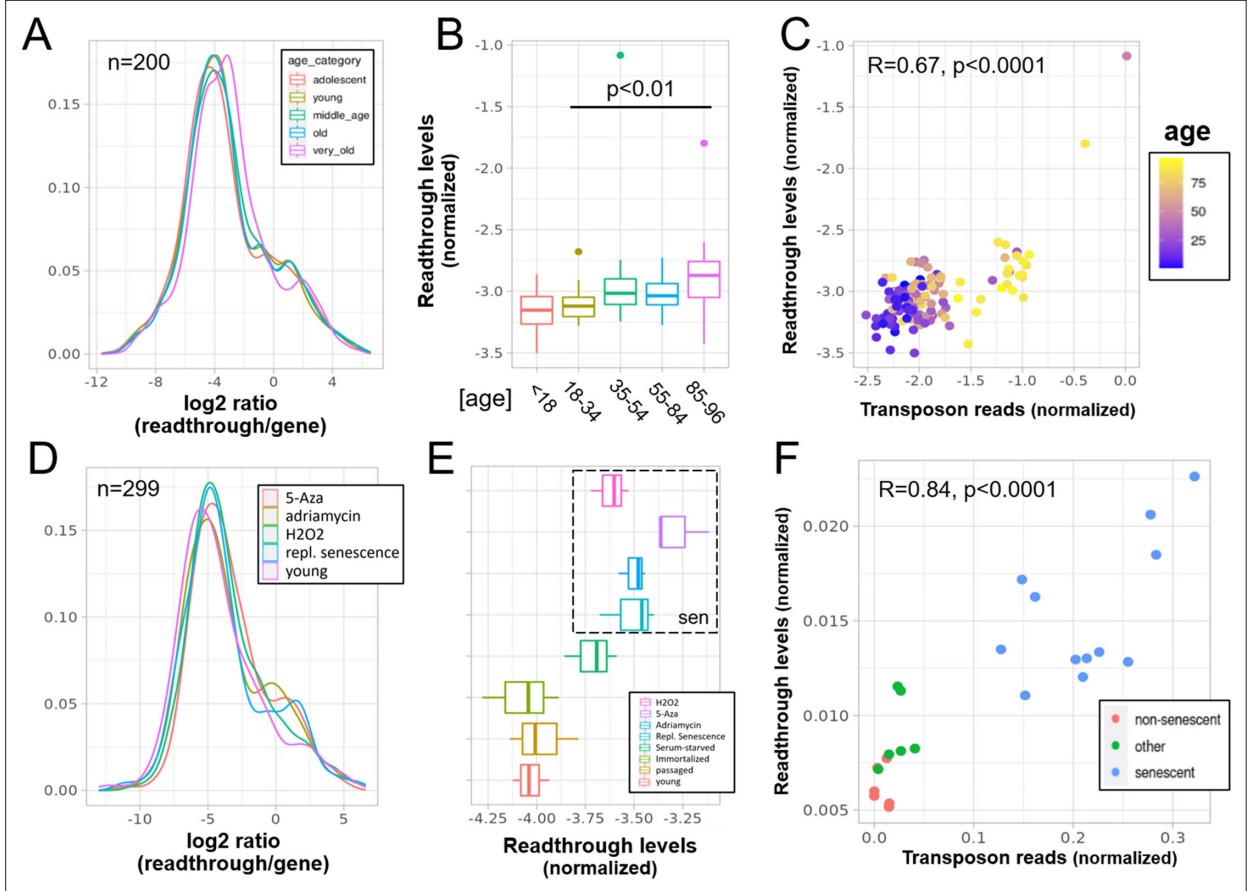

**Figure 4.** Increased readthrough levels with aging and senescence. Readthrough transcription is increased in fibroblasts isolated from the very old (**A, B**) and after induction of senescence in vitro (**D, E**). Readthrough transcription is correlated with transposon expression (**C, F**). (**A**) Readthrough was determined in a region 10- to 20-kb downstream of genes for a subset of genes that were at least 20 kb away from the nearest neighboring gene (n = 200 regions). The log2 ratio of readthrough to gene expression is plotted across five age groups (adolescent n = 32, young n = 31, middle-aged n = 22, old n = 37, and very old n = 21). (**B**) As in (**A**), but data are plotted on a per sample basis. (**C**) Scatterplot between transposon expression and readthrough levels (normalized as in A and *Figure 1*) for all 143 samples. Each sample is colored by age. (**D**) Readthrough was determined in a region 10- to 20-kb downstream of genes for a subset of genes that were at least 20 kb away from the nearest neighboring gene (n = 299 regions). The log2 ratio of readthrough to gene expression is plotted across four senescent conditions (H$_2$O$_2$, 5-Aza, adriamycin, and replicative senescence) and for early passage cells. N = 3 per group. (**E**) As in (**D**), but data are plotted on a per sample basis and for additional control datasets (serum-starved, immortalized, intermediate passage, and early passage). N = 3 per group. (**F**) Scatterplot between transposon expression and readthrough levels (normalized as in A and *Figure 1*) for all 24 samples. Each sample is colored by senescence status.

The online version of this article includes the following figure supplement(s) for figure 4:

**Figure supplement 1.** Readthrough transcription might be increased with age in mouse liver, but not during in vitro senescence of mouse fibroblasts.

**Figure supplement 2.** Readthrough is increased downstream of genes over a wide range of distances.

**Figure supplement 3.** Read-in does not change with age and is increased by senescence.

**Figure supplement 4.** Readthrough and transposon counts are higher closer to genes.

**Figure supplement 5.** Age-related transposon expression downstream of high readthrough genes is elevated.

**Figure supplement 6.** Intergenic transposons are rarely expressed.

**Figure supplement 7.** Age-related changes in readthrough are correlated with changes in transposon expression.

**Figure supplement 8.** Positive correlation between readthrough counts and transposons downstream of genes.

of the same genes (*Figure 4—figure supplement 7*). Transposons located elsewhere showed a weaker correlation with readthrough.

To further test whether these correlations between log-fold changes are also reflected on the count level, we determined the correlation between readthrough counts and the counts of transposons downstream of genes for each sample individually. Transposons located upstream of genes

and randomized transposons served as a negative control. Consistent with the above data, we found a specific correlation between transposon and readthrough counts in both datasets (*Figure 4—figure supplement 8*).

Finally, we asked whether readthrough levels are predictive of transposon expression on a per sample basis independently of intron retention. Using the aging dataset with 143 samples we compared a linear model just including intron retention with one also including readthrough and found the second model to perform significantly better (p < 0.0001). In addition, we found that the correlation between readthrough levels and transposon expression remained significant even when correcting for intron retention using partial correlation (R = 0.53, p < 0.0001). We did not test this in the senescence dataset due to its limited size.

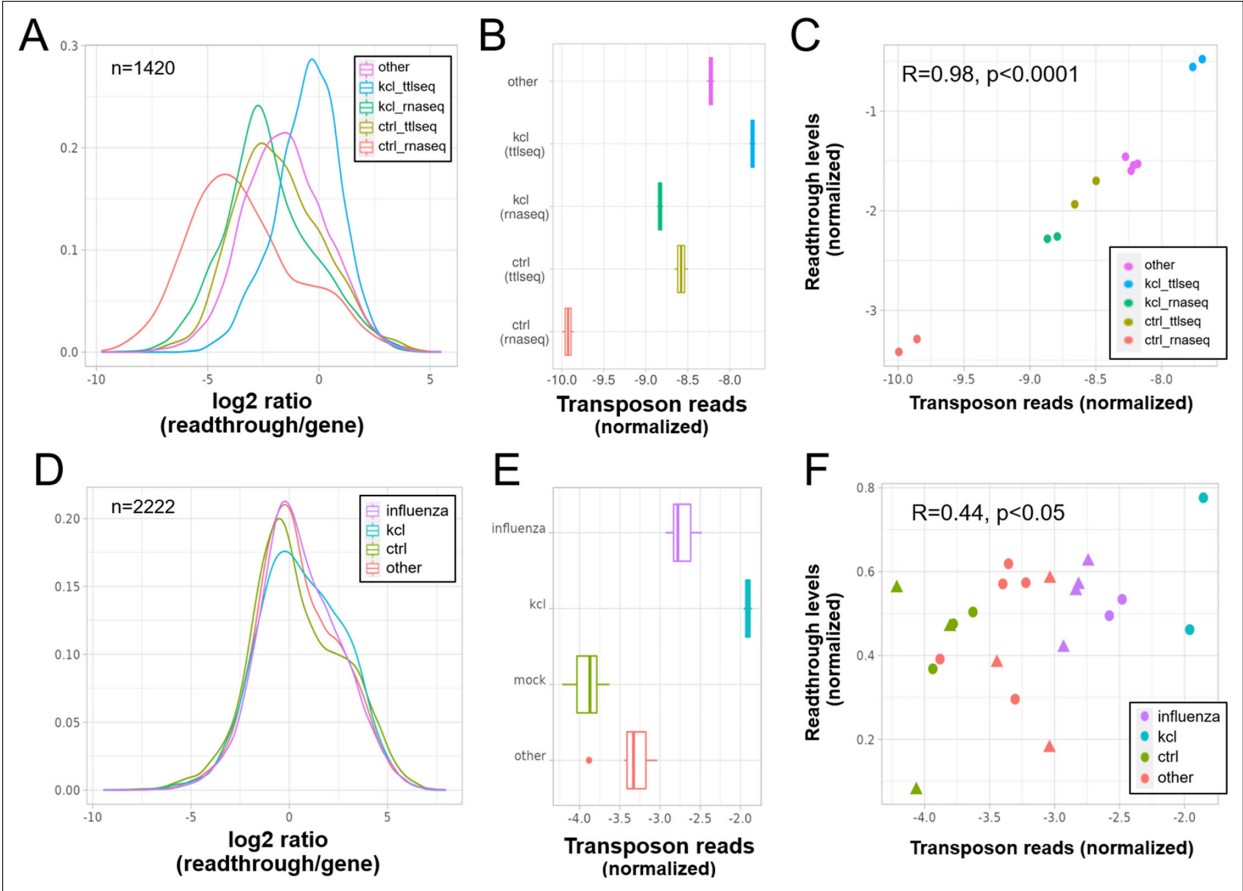

**Figure 5.** Elevated transposon expression after induced readthrough. Readthrough transcription is increased after KCl treatment of HEK293 cells (**A, D**) and to a lesser extent during influenza infection (**D**). These readthrough inducing treatments also promote increased transposon expression (**B, E**). Giving rise to a correlation between readthrough and transposon expression on a per sample basis (**C, F**). Data in (**A–C**) are from *Rosa-Mercado et al., 2021*. In this study, HEK293 cells were subjected to 1 hr of hyperosmotic stress (KCl) after which RNA-seq and TT-TL-seq were performed. Data in (**D–F**) are from *Bauer et al., 2018*. In this study, HEK293 cells were subjected to 1 hr of hyperosmotic stress (KCl) after which mNETseq was performed. (**A**) Readthrough was determined in a region 10- to 20-kb downstream of genes for a subset of genes that was at least 20 kb away from the nearest neighboring gene (*n* = 1420 regions). (**B**) To normalize transposon expression counts for each transposon were corrected for the expression of the nearest gene. Normalized transposon counts for each sample are shown as box-whisker plot. (**C**) Normalized transposon counts (as in B) and readthrough levels are plotted for each sample (*n* = 12). (**D**) Readthrough was determined in a region 10- to 20-kb downstream of genes for a subset of genes that was at least 20 kb away from the nearest neighboring gene (*n* = 2222 regions). (**E**) To normalize transposon expression counts for each transposon were corrected for the expression of the nearest gene. Normalized transposon levels for each sample are shown as box-whisker plot. (**F**) Normalized transposon counts (as in B) and readthrough counts are plotted for each sample (*n* = 22). The shapes indicate the cell type (circles = HEK293, triangles = A549).

The online version of this article includes the following figure supplement(s) for figure 5:

**Figure supplement 1.** Elevated transposon expression after induced readthrough.

**Figure supplement 2.** Heatshock and osmotic stress promote expression of gene proximal transposons.

## Readthrough is a causal inducer of transposon expression

Based on this correlative evidence (*Figure 4*), we hypothesized that specific inducers of readthrough would also increase transposon expression. To test this, we reanalyzed literature data from studies involving multiple treatments that promote readthrough. These included viral infection, hyperosmotic stress (KCl treatment), heat shock, and oxidative stress.

We found that KCl treatment of HEK293 cells for 1 hr (*Rosa-Mercado et al., 2021*) indeed strongly induced both readthrough (*Figure 5A*) and transposon expression (*Figure 5B*) as determined by RNA-seq and TTLseq. Consistent with this, in our reanalysis of a study using mNET-seq profiling of nascent mRNA (*Bauer et al., 2018*), KCl treatment of HEK293 cells also induced readthrough (*Figure 5D*) and transposon expression (*Figure 5E*), whereas influenza infection caused an intermediate phenotype. Finally, we analyzed RNA-seq data from NIH-3T3 mouse fibroblasts subjected to three stressors over 2 hr (*Vilborg et al., 2017*). Heatshock caused extensive readthrough (*Figure 5—figure supplement 1A*) and transposon expression (*Figure 5—figure supplement 1B*), whereas hydrogen peroxide treatment and KCl caused moderate expression of these.

We also reasoned that readthrough should specifically increase the expression of transposons that are close to genes. Consistent with this notion, induced readthrough led to preferential expression of gene proximal transposons (i.e. those within 25 kb of genes), when compared with senescence or

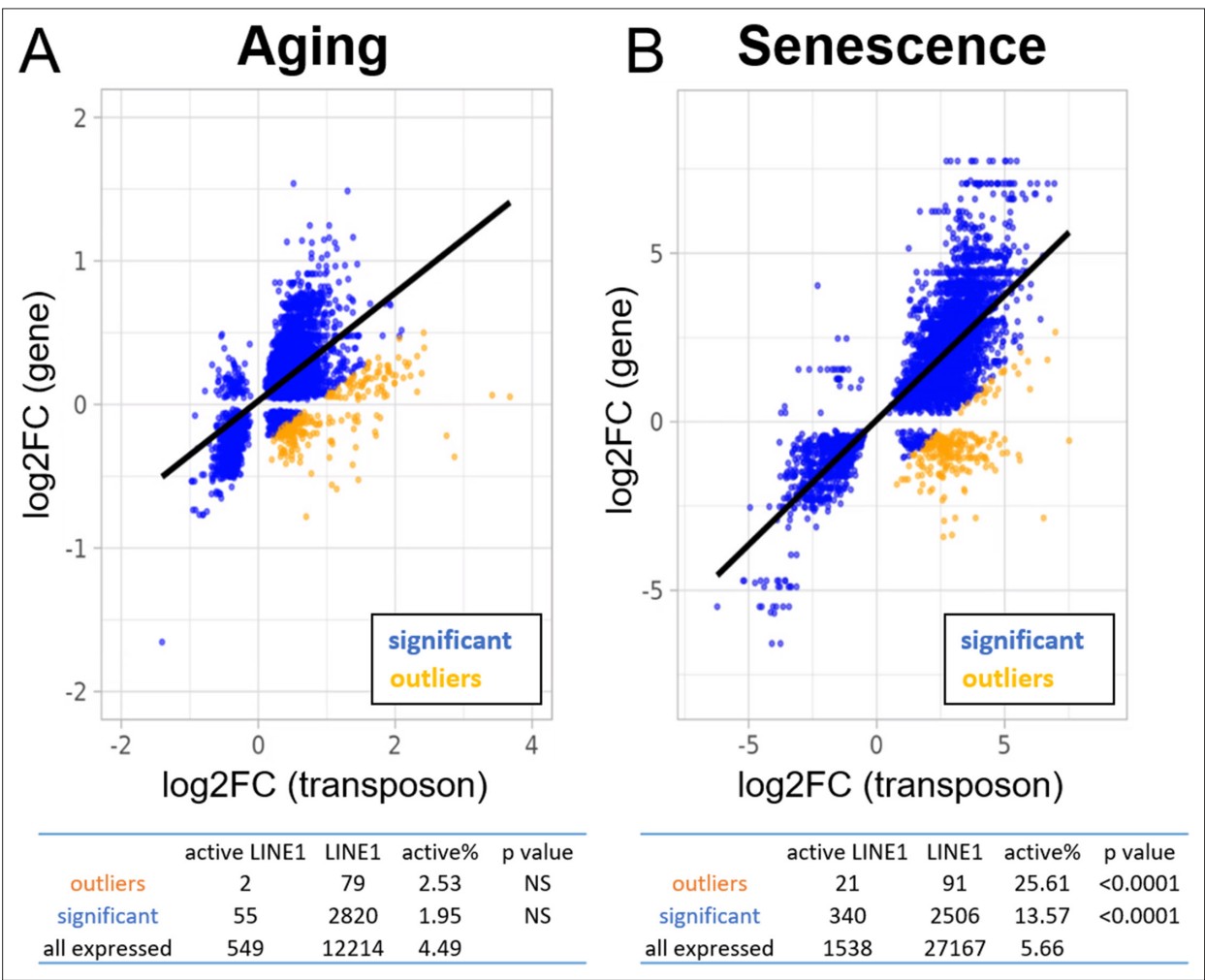

**Figure 6.** Increased expression of active LINE-1 elements with senescence but not aging. A scatterplot between the log2-fold change of every transposon (p < 0.05) differentially expressed with age or senescence and the log2-fold change of the nearest gene. The percentage of active LINE-1s among transposons with higher-than-expected expression levels is unchanged in aging (**A**) and increased during senescence (**B**). For the tables, we compared the percentage of active LINE-1s among all LINE-1 elements for 250 transposons with higher-than-expected expression levels (in orange; labeled 'outliers'), all significantly changed transposons (blue plus orange; labeled 'significant'), and all transposons passing read filtering (not plotted; labeled 'all expressed'). The p-value is based on the comparison with 'all expressed'.

aging (*Figure 5—figure supplement 2A*). Moreover, the fraction of significantly upregulated transposons found outside of genes was higher with induced readthrough than with senescence or aging (*Figure 5—figure supplement 2B*).

## Increased expression of autonomous LINE-1 elements with senescence but not aging

Many expressed LINE-1 elements are found within genes, which poses a challenge for the quantification of active LINE-1s. Given this proximity it is unsurprising that a substantial correlation between the expression of transposons and neighboring genes (*Figure 6*) is seen in RNA-seq.

To mitigate this problem, we selected 250 transposons that showed higher than expected expression levels compared to the nearest gene. We reasoned that LINE-1s with a functional promoter ('active' LINE-1s) would be more likely autonomous and should be enriched among transposons that show a divergent expression profile from their neighboring genes.

We found no enrichment for active LINE-1 elements in the aging dataset (*Figure 6A*). In contrast, transposons significantly upregulated with senescence and, among those, especially the ones with higher-than-expected expression, showed significant enrichment for active LINE-1s (*Figure 6B*). Moreover, we found that almost 100% of active LINE-1s were increased with senescence which was higher than for all LINE-1s and other transposons.

Finally, we performed the same analysis with a different definition of active LINE-1s as those potentially encoding functional ORF1p and ORF2p. In this analysis, we included all elements encoding an open reading frame >900 nt with high sequence similarity to ORF1p and >3400 nt with high sequence similarity to ORF2p (p < 0.05 for pairwise alignment). Out of 122,626 annotated LINE-1s 585 encoded a putative ORF1p and 133 encoded an ORF2p.

ORF1p-encoding LINE-1s were significantly enriched among transposons with higher-than-expected expression and among transposons significantly changed with senescence or aging. These findings were more robust for senescence than for aging. ORF2p-encoding LINE-1s, in contrast, were too rare to be studied reliably as to their enrichment.

## Discussion

We show that transposons would serve as a useful biomarker of aging, even if most of the transposon signal in RNA-seq was due to co-expression with nearby transcriptional units, because they capture different kinds of transcriptional dysfunction.

### Introns

Intron retention is one prominent example of dysfunctional splicing during aging, but the association with age-related transposon expression has not been evaluated. A recent study by *Gualandi et al., 2022* using RNA-seq data from the 1000 Genomes project found a strong per sample correlation between intron and transposon expression in the general population. In line with these results, we showed that intron retention may explain a large fraction of transposon reads in aging and senescence datasets (*Figure 1*).

### Readthrough, read-in, and transposons

In a study of oncogene-induced senescence, *Muniz et al., 2017* found increased readthrough transcription at 91 so-called START loci, which was associated with suppression of neighboring genes through the production of antisense RNAs and several of these antisense RNAs were necessary for the maintenance of the senescent state. Our data are consistent with their model, although we find even more extensive readthrough transcription. Analyzing hundreds of loci we show that senescence promotes transcriptional readthrough leading to the overexpression of intergenic transposons (>25 kb from the nearest gene) and read-in at neighboring genes (*Figure 3B*, *Figure 4—figure supplement 3B*). In contrast, aging-related readthrough was qualitatively different, producing shorter readthrough and no read-in (*Figure 3B*, *Figure 4—figure supplement 3A*).

Our correlative data suggested that readthrough transcription could account for the expression of many transposons that are outside of genes and thus cannot be explained by intron retention (*Figure 4C, F*). To test this hypothesis, we reanalyzed data from models of induced readthrough

showing that readthrough is indeed a causal inducer for the expression of gene proximal transposons after hyperosmotic stress and viral infection (*Figure 5*). Given the technical challenges of measuring readthrough and the unreliability of intron retention detection tools (*David et al., 2022*), our results are likely underestimating the true magnitude of correlation between readthrough and transposon expression.

In a previous study, it was observed that senescence leads to increased expression of transposons and of genes with low abundances (*Zhang et al., 2021*). This 'leakage' was attributed to loss of H3K9me3 histone marks and increased heterochromatin accessibility, but is also consistent with readthrough transcription, as we observed. Indeed, it has been suggested that readthrough is the most likely cause for apparent intergenic transcription (*Agostini et al., 2021*).

## Is there genuine expression of autonomous transposons during aging?

The distribution of transposons showing apparent age-related expression within the genome is an important question since intragenic transposons are more likely to be co-expressed with genes. Using Sanger sequencing of long-range amplicons to study replicative senescence, *De Cecco et al., 2019* found a large fraction of transposons mapping to intergenic (69%) regions and few to intragenic/gene proximal regions (31%). In contrast using RNA-seq data, we find that most transposons are located within genes (≥70%; *Table 1*), although, there are more intergenic transposons significantly changed with senescence than with aging (*Figure 3*). One explanation for this discrepancy is the mapping approach used. Since their analysis did not use an expectation–maximization algorithm to map back transposons onto the genome, it may underestimate the number of intragenic and gene-adjacent transposons in contrast to our analysis using TELocal.

Given that inactive LINE-1 loci greatly outnumber active loci (*Deininger et al., 2017*; *Stow et al., 2021*), and we saw only limited evidence for active LINE-1 transcription in fibroblasts isolated from donors of varying ages (*Figure 6A*), our data appear to question the role of transposons in organismal aging. An intriguing explanation is provided by *De Cecco et al., 2019*, who found that LINE-1 ORF1 protein-positive cells colocalize with a small subset of senescent cells. Most likely, such changes would not be obvious from bulk sequencing, perhaps explaining why we did not see elevated LINE-1 expression in the aging dataset. In contrast, the signal would be readily detectable in cultures induced into senescence where most cells do enter senescence, which is consistent with our findings (*Figure 6B*).

As for another explanation, we speculate that LINE-1 elements expressed through intron retention and readthrough are compatible with translation via leaky scanning or re-initiation (*Mouilleron et al., 2016*), accounting for increased LINE-1 ORF1 protein levels detected in senescent cells, even if leaky scanning is orders of magnitude less efficient than normal initiation.

We did not study the expression of LTR or SINE elements in detail, which have been implicated in organismal aging (*Liu et al., 2023*) and age-related diseases such as macular degeneration (*de Koning et al., 2011*), because we lack consensus annotations for active and inactive members of these families.

## Summary and limitations

Taken together, our work and existing literature suggest that RNA-seq, and by extension qPCR, of transposable elements overestimates their true expression and their age-related changes. Such data alone are insufficient to support a 'transposon hypothesis of aging' that postulates an increase in autonomous expression of such elements. Instead, a concerted increase in transposon expression, intron retention, and transcriptional readthrough points to a global increase in transcriptional dysfunction during aging and senescence, at least in human cells. Nevertheless, there are several other techniques such as immunoblotting for ORF1 protein, rapid amplification of cDNA ends (RACE), or genomic PCR that may provide evidence in favor of age-related reactivation of potentially autonomous transposons, specifically LINE-1 elements (*De Cecco et al., 2019*).

Furthermore, since most of our analysis focused on cellular models of aging and senescence, we cannot exclude that RNA-seq data performed on tissues samples would more faithfully and specifically capture the expression of autonomous transposable elements.

Although we have shown increased transcriptional dysfunction with aging and cellular senescence, so far we lack a model to explain these findings. In this context, it is noteworthy that the elongation rate of Pol II is correlated with readthrough transcription (*Fong et al., 2015*), intron processing, and

may increase with aging (*Debès et al., 2023*), potentially providing a unifying explanation. Another potential explanation that could account for both elevated readthrough and expression of intergenic transposon is loss of repressive heterochromatin (*Zhang et al., 2021*) during aging or senescence. We were unable to test this hypothesis directly since we only had access to RNA-seq data for these samples.

## Methods

### Dataset selection, read preprocessing and alignment

We searched the SRA for suitable datasets that used total RNA extraction because this improves the detection of non-canonical transcripts that may be degraded before successful polyadenylation. Publicly available human RNA-seq datasets of aging and senescence, respectively, were obtained from *Colombo et al., 2018* (GSE60340) and *Fleischer et al., 2018* (GSE113957). Human RNA-seq and TT-TimeLapse-seq (TT-TL-seq) data were obtained from *Rosa-Mercado et al., 2021* (GSE152059), human native elongating transcript sequencing (mNET-seq) data from *Bauer et al., 2018* (PRJNA432639), and mouse RNA-seq data from *Vilborg et al., 2017* (GSE98906).

Mouse liver RNA-seq data from ad libitum and dietary restricted mice were obtained from *Hahn et al., 2017* (GSE92486) and mouse replicative senescence data from *Wang et al., 2022* (GSE179880).

Reads were filtered, adaptor-trimmed and repaired using fastp (*Chen et al., 2018*) with a min Phred score of 25. Afterwards reads passing these filters were aligned to the human genome (GRCh38) or mouse genome (GRCm39) using STAR 2.7.8a (*Dobin et al., 2013*) in single-pass mode. To optimize the quantification of transposons the STAR parameters `--outFilterMultimapNmax` and `--winAnchorMultimapNmax` were set to 100. The full pipeline with all the configuration files is available on github (https://github.com/pabisk/aging_transposons copy archived at *Pabis, 2024*).

### Detection of readthrough and intron retention

We employed a modified Automatic Readthrough Transcription Detection (ARTDeco) pipeline, using their provided annotations (*Roth et al., 2020*) but with a different quantification approach. Readthrough candidate regions in the ARTDeco pipeline were defined based on the longest transcript to exclude the effects of differential 3'-UTR usage. Genic and readthrough reads were counted using featureCounts from the Rsubread package in 10 kb bins downstream of genes, in a window between 0 and 100 kb. Readthrough regions with low coverage, in close proximity to, or overlapping, nearby protein-coding genes were excluded from the analysis. Intron retention was detected using the intron REtention Analysis and Detector (iREAD), which is a tool that detects retention events based on both splice junction reads and intron expression levels (*Li et al., 2020*). All samples were treated as unstranded.

### Locus-specific detection of transposon reads and transcripts

We used TELocal v1.1.1 (mhammell-laboratory/TElocal) to quantify transposon reads, which is functionally similar to TETranscripts (*Jin et al., 2015*) except it provides locus-specific expression levels. Both tools use an expectation maximization algorithm to assign ambiguous reads. Annotation files were downloaded from the mhammell lab and from genecode (v39 for human and M27 for mouse). TELocal was also used to quantify transcriptomic counts. Again, all samples were treated as unstranded.

### Expression and normalization of genes, introns, transposons, and readthrough transcripts

Transcripts and elements with low expression counts were filtered. After filtering, intron, readthrough, read-in, or transposon elements were combined with transcript counts for the analysis in DESeq2. For the aging fibroblast dataset age was normalized and centered and the design model considered age + gender + ethnicity. For the induced senescence dataset all four inducers were included together and the remaining treatment groups were considered non-senescent.

In order to study expression of these elements unbiased by gene co-expression, we selected the top 1000 overexpressed elements and divided their read counts by the read counts of the nearest gene. When not specified in the annotation the nearest genes or elements were identified with the

bedtools closest or intersect function (*Quinlan and Hall, 2010*), as appropriate. Finally, to provide a single expression value for each sample we used the mean log of the normalized expression values.

## Expression of potentially active and protein-coding LINE-1 elements

We defined potentially active LINE-1 elements as those that have a functional promoter (*McKerrow and Fenyö, 2020*). Whereas to define potentially protein-coding LINE-1 elements we predicted open reading frames for each annotated element using the predORF function from the systemPipeR package. Global pairwise alignment to ORF1p (uniprot: Q9UN81) and ORF2p (uniprot: O00370) was performed and only elements with a significant alignment were included for further analyses (p < 0.05).

## Acknowledgements

We would like to thank Juliane Liepe, Marco Malavolta, and Chin-Tong Ong for support and/or their help in improving this manuscript and VitaDAO for financial support.

## Additional information

### Funding

| Funder | Grant reference number | Author |
|---|---|---|
| National University of Singapore | NUHSRO/2020/114 | Brian K Kennedy |

The funders had no role in study design, data collection, and interpretation, or the decision to submit the work for publication.

### Author contributions

Kamil Pabis, Conceptualization, Formal analysis, Investigation, Methodology, Writing - original draft; Diogo Barardo, Conceptualization, Writing - review and editing; Olga Sirbu, Methodology, Writing - review and editing; Kumar Selvarajoo, Conceptualization, Supervision, Methodology, Writing - review and editing; Jan Gruber, Formal analysis, Supervision, Writing - review and editing; Brian K Kennedy, Conceptualization, Supervision, Writing - review and editing

### Author ORCIDs

Kamil Pabis http://orcid.org/0000-0002-1120-3079
Jan Gruber http://orcid.org/0000-0003-3329-3789
Brian K Kennedy http://orcid.org/0000-0002-5754-1874

Joint Public Review: https://doi.org/10.7554/eLife.87811.3.sa1
Author Response https://doi.org/10.7554/eLife.87811.3.sa2

## Additional files

### Supplementary files
• MDAR checklist
• Supplementary file 1. Supplementary figures and tables.

### Data availability

The current manuscript is a computational study, so no data have been generated for this manuscript.

The following previously published datasets were used:

| Author(s) | Year | Dataset title | Dataset URL | Database and Identifier |
|---|---|---|---|---|
| Purcell M, Tainsky M, Kruger A | 2015 | Pathway Profiling of Replicative and Induced Senescence | https://www.ncbi.nlm.nih.gov/geo/query/acc.cgi?acc=GSE60340 | NCBI Gene Expression Omnibus, GSE60340 |
| Fleisher JG, Schulte R, Tsai H, Tyagi S, Ibarra A, Shokhirev MN, Huang L, Hetzer MW, Navlakha S | 2018 | Predicting age from the transcriptome of human dermal fibroblasts | https://www.ncbi.nlm.nih.gov/geo/query/acc.cgi?acc=GSE113957 | NCBI Gene Expression Omnibus, GSE113957 |
| Rosa-Mercado NA, Zimmer JT, Apostolidi M, Rinehart J, Simon MD, Steitz JA | 2021 | TT-TL-seq reveals transcriptional profiles that accompany DoG induction after hyperosmotic stress. | https://www.ncbi.nlm.nih.gov/geo/query/acc.cgi?acc=GSE152059 | NCBI Gene Expression Omnibus, GSE152059 |
| Bauer et al. | 2018 | Meromictic Lake Metagenome | https://www.ncbi.nlm.nih.gov/bioproject/?term=PRJNA257655 | NCBI BioProject, PRJNA257655 |
| Vilborg A, Shalgi R | 2017 | Comparative analysis reveals genomic features of stress-induced transcriptional readthrough | https://www.ncbi.nlm.nih.gov/geo/query/acc.cgi?acc=GSE98906 | NCBI Gene Expression Omnibus, GSE98906 |
| Hahn et al. | 2017 | Dietary restriction protects from age-associated DNA methylation and induces epigenetic reprogramming of lipid metabolism | https://www.ncbi.nlm.nih.gov/geo/query/acc.cgi?acc=GSE92486 | NCBI Gene Expression Omnibus, GSE92486 |
| Wang Y, Liu L, Song Y | 2022 | Unveiling E2F4, TEAD1 and AP-1 as regulatory transcription factors of the replicative senescence program by multi-omics analysis | https://www.ncbi.nlm.nih.gov/geo/query/acc.cgi?acc=GSE179880 | NCBI Gene Expression Omnibus, GSE179880 |

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
