## [Editor Report · eLife assessment]

This study presents **fundamental** findings on the role of transcription readout and intron retention in transposon expression during aging in mammals. The evidence supporting the claims of the authors is **compelling**, strongly supporting the authors' claims. The work will be of interest to scientists studying aging, transcription regulation, and epigenetics.

---

## [Referee Report · Joint Public Review]

In this manuscript, the authors examined the role of transcription readout and intron retention in increasing transcription of transposable elements during aging in mammals. It is assumed that most transposable elements have lost the regulatory elements necessary for transcription activation. Using available RNA-seq datasets, the authors showed that an increase in intron retention and readthrough transcription during aging contributes to an increase in the number of transcripts containing transposable elements.

Previously, it was assumed that the activation of transposable elements during aging is a consequence of a gradual imbalance of transcriptional repression and a decrease in the functionality of heterochromatin (de repression of transcription in heterochromatin). Therefore, this is an interesting study with important novel conclusion.

The authors revised the manuscript in accordance with the comments. Overall, the manuscript is useful because it shows that there is no direct connection between increased levels of transposon RNA and aging, and further demonstrates the disorganization of the transcriptional apparatus during aging.

---

## [Author Response]

The following is the authors’ response to the original reviews.

**Public Reviews:**

**Reviewer #1 (Public Review):**
Transcriptional readthrough, intron retention, and transposon expression have been previously shown to be elevated in mammalian aging and senescence by multiple studies. The current manuscript claims that the increased intron retention and readthrough could completely explain the findings of elevated transposon expression seen in these conditions. To that end, they analyze multiple RNA-seq expression datasets of human aging, human senescence, and mouse aging, and establish a series of correlations between the overall expression of these three entities in all datasets.While the findings are useful, the strength of the evidence is incomplete, as the individual analyses unfortunately do not support the claims. Specifically, to establish this claim there is a burden of proof on the authors to analyze both intron-by-intron and gene-by-gene, using internal matched regions, and, in addition, thoroughly quantify the extent of transcription of completely intergenic transposons and show that they do not contribute to the increase in aging/senescence. Furthermore, the authors chose to analyze the datasets as unstranded, even though strand information is crucial to their claim, as both introns and readthrough are stranded, and if there is causality, than opposite strand transposons should show no preferential increase in aging/senescence. Finally, there are some unclear figures that do not seem to show what the authors claim. Overall, the study is not convincing.Major concerns:1. Why were all datasets treated as unstanded? Strand information seems critical, and should not be discarded. Specifically, stranded information is crucial to increase the confidence in the causality claimed by the authors, since readthrough and intron retention are both strand specific, and therefore should influence only the same strand transposons and not the opposite-strand ones.

This is an excellent suggestion. Since only one of our datasets was stranded, we did not run stranded analyses for the sake of consistency. We would like to provide two analyses here that consider strandedness:

First, we find that within the set of all expressed transposons (passing minimal read filtering), 86% of intronic transposons match the strand of the intron (3147 out of 3613). In contrast, the number is 51% after permutation of the strands. Similarly, when we randomly select 1000 intronic transposons 45% match the strandedness of the intron (here we select from the set of all transposons). This is consistent with the idea that most transposons are only detectable because they are co-expressed on the sense strand of other features that are highly expressed.

As for the readthrough data, 287 out of 360 transposons (79%) within readthrough regions matched the strand of the gene and its readthrough.

Second, in the model we postulate, the majority of transposon transcription occurs as a co-transcriptional artifact. This applies equally to genic transposons (gene expression), intronic (intron retention) and gene proximal (readthrough or readin) transposons. Therefore, we performed the following analysis for the set of all transposons in the Fleischer et al. fibroblast dataset.

When we invert the strand annotation for transposons, before counting and differential expression, we would expect the counts and log fold changes to be lower compared to using the “correct” annotation file.

Indeed, we show that out of 6623 significantly changed transposons with age only 226 show any expression in the “inverted run” (-96%). (Any expression is defined as passing basic read filtering.)

Out of the 226 transposons that can be detected in both runs most show lower counts (A) and age-related differential expression converging towards zero (B) in the inverted run (Fig. L1).

**Author response image 1. sa2fig1:** Transposons with inverted strandedness (“reverse”) show lower expression levels (log counts; A) and no differential expression with age (B) when compared to matched differentially expressed transposons (“actual”). For this analysis we selected all transposons showing significant differential expression with age in the actual dataset that also showed at least minimal expression in the strand-inverted analysis (n=226). Data from Fleischer et al. (2018). (A) The log (counts) are clipped because we only used transposons that passed minimal read filtering in this analysis. (B) The distribution of expression values in the actual dataset is bimodal and positive since some transposons are significantly up- or downregulated. This bimodal distribution is lost in the strand-inverted analysis.

1. "Altogether this data suggests that intron retention contributes to the age-related increase in the expression of transposons" - this analysis doesn't demonstrate the claim. In order to prove this they need to show that transposons that are independent of introns are either negligible, or non-changing with age.

We would like to emphasize that we never claimed that intron retention and readthrough can explain all of the age-related increases in transposon expression. In fact, our data is compatible with a multifactorial origin of transposons expression. Age- and senescence-related transposon expression can occur due to: 1/ intron retention, 2/ readthrough, 3/ loss of intergenic heterochromatin. Specifically, we do not try to refute 3.

However, since most transposons are found in introns or downstream of genes, this suggests that intron retention and readthrough will be major, albeit non-exclusive, drivers of age-related changes in transposons expression. Even if the fold-change for intergenic transposons with aging or senescence were higher this would not account for the broadscale expression patterns seen in RNAseq data.

To further illustrate this, we analyzed transposons located in introns, genes, downstream (ds) or upstream (us) of genes (distance to gene < 25 kb) or in intergenic regions (distance to gene > 25 kb). Indeed, we find that although intergenic transposons show similar log-fold changes to other transposon classes (Fig. L2A), their total contribution to read counts is negligible (Fig. L2B, Fig. Fig. S15). We have also now added a more nuanced explanation of this issue to the discussion.

**Author response image 2. sa2fig2:** We analyzed transposons located in introns, genes, downstream (ds) or upstream (us) of genes (distance to gene < 25 kb) or in intergenic regions (distance to gene > 25 kb). Independent of their location, transposons show similar differential expression with aging or cellular senescence (A). In contrast, the expression of transposons (log counts) is highly dependent on their location and the median log(count) value decreases in the order: genic > intronic > ds > us > intergenic.

**Author response image 3. sa2fig3:** Total counts are the sum of all counts from transposons located in introns, genes, downstream (ds) or upstream (us) of genes (distance to gene < 25 kb) or in intergenic regions (distance to gene > 25 kb). Counts were defined as cumulative counts across all samples.

1. Additionally, the correct control regions should be intronic regions other than the transposon, which overall contributed to the read counts of the intron.1. Furthermore, analysis of read spanning intron and partly transposons should more directly show this contribution.

Thank you for this comment. To rephrase this, if we understand correctly, the concern is that an increase in transposon expression could bias the analysis of intron retention since transposons often make up a substantial portion of an intron. We would like to address this concern with the following three points:

First, if the concern is the correlation between log fold-change of transposons vs log fold-change of their containing introns, we do not think that this kind of data is biased. While transposons make up much of the intron, a single transposon on average only accounts for less than 10% of an intron.

Second, to address this more directly, we show here that even introns that do not contain expressed transposons are increased in aging fibroblasts and after induction of cellular senescence (Fig. S8). This shows that intron retention is universal and most likely not heavily biased by the presence or absence of expressed transposons.

**Author response image 4. sa2fig4:** We split the set of introns that significantly change with cellular aging (A) or cell senescence (B) into introns that contain at least one transposon (has_t) and those that do not contain any transposons (has_no_t). Intron retention is increased in both groups. In this analysis we included all transposons that passed minimal read filtering (n=63782 in A and n=124173 in B). Median log-fold change indicated with a dashed red line for the group of introns without transposons.

Third, we provide an argument based on the distribution of transposons within introns (Fig. L3).

**Author response image 5.**

The 5’ and 3’ splice sites show the highest sequence conservation between introns, whereas the majority of the intronic sequence does not. This is because these sites contain binding sites for splicing factors such as U1, U2 and SF1 (A). Transposons could affect splicing and we present a biologically plausible mechanism and two ancillary hypotheses here (B). If transposons affect the splicing (retention) of introns the most likely mechanism would be via impairment of splice site recognition because a transposon close to the site forms a secondary structure, binds an effector protein or provides inadequate sequences for pairing. Hypothesis 1: Transposons impair splicing because they are close to the splice site. Hypothesis 2: Transposons do not impair splicing because they are located away from the splice junction. Retained introns should show a similar depletion of transposons around the junction.

Image adapted from: Ren, Pingping, et al. "Alternative splicing: a new cause and potential therapeutic target in autoimmune disease." Frontiers in Immunology 12 (2021): 713540.

Consistent with hypothesis 2 (“transposons do not impair splicing”), we show that the distribution of transposons within introns is similar for the set of all transposons and all significant transposons within significantly overexpressed introns (Fig. S7. A and B is similar in the case of aged fibroblasts; D and E is similar in the case of cellular senescence). If transposon expression was causally linked to changes in intron retention, the most likely mechanism would be via an impairment of splicing. We would expect transposons to be located close to the splice junction, which is not what we observed. Instead, the data is more consistent with intron retention as a driver of transposon expression.

**Author response image 6. sa2fig5:** Transposons are evenly distributed within introns except for the region close to splice junctions (**A-E**). Transposons appear to be excluded from the splice junction-adjacent region both in all introns (**A, D**) and in significantly retained introns (**B, E**). In addition, transposon density of all introns and significantly retained introns is comparable (**C, F**). We included only introns containing at least one transposon in this analysis. (**A**) Distribution of 2292769 transposons within 163498 introns among all annotated transposons. (**B**) Distribution of 195190 transposons within 14100 introns significantly retained with age. (**C**) Density (transposon/1kb of intron) of transposons in all introns (n=163498) compared to significantly retained introns (n=14100). (**D**) as in (**A**) (**E**) Distribution of 428130 transposons within 13205 introns significantly retained with induced senescence. (**F**) Density (transposon/1kb of intron) of transposons in all introns (n=163498) compared to significantly retained introns (n=13205).

1. "This contrasts with the almost completely even distribution of randomly permuted transposons." How was random permutation of transposons performed? Why is this contract not trivial, and why is this a good control?

Permutation was performed using the bedtools shuffle function (Quinlan et al. 2010). We use the set of all annotated transposons and all reshuffled transposons as a control. It is interesting to observe that these two show a very similar distribution with transposons evenly spread out relative to genes. In contrast, expressed transposons are found to cluster downstream of genes. This gave rise to our initial working hypothesis that readthrough should affect transposon expression.

1. Fig 4: the choice to analyze only the 10kb-20kb region downstream to TSE for readthrough regions has probably reduced the number of regions substantially (there are only 200 left) and to what extent this faithfully represent the overall trend is unclear at this point.

This is addressed in Suppl. Fig. 7, we repeated the analysis for every 10kb region between 0 and 100kb, showing similar results.

Furthermore, we show below in a new figure that the results are comparable when we measure readthrough in the 0 to 10kb region, while the sample size of readthrough regions is increased.

Finally, it is commonly accepted to remove readthrough regions overlapping genes, which while reducing sample size, increases accuracy for readthrough determination (Rosa-Mercado et al. 2021). Without filtering readthrough regions can overlap neighboring genes which is reflected in an elevated ratio of Readthrough_counts/Genic_counts (Fig. S9).

**Author response image 7. sa2fig6:** Readthrough was determined in a region 0 to 10 kb downstream of genes for a subset of genes that were at least 10 kb away from the nearest neighboring gene (n=684 regions). The log2 ratio of readthrough to gene expression is plotted across five age groups (adolescent n=32, young n=31, middle-aged n=22, old n=37 and very old n=21). (**B**) As in (**A**) but data is plotted on a per sample basis. (**C**) Readthrough was determined in a region 0 to 10 kb downstream of genes for a subset of genes that were at least 10 kb away from the nearest neighboring gene (n=1045 regions). The log2 ratio of readthrough to gene expression is plotted for the groups comprising senescence (n=12) and the non-senescent group (n=6). (**D**) As in (**D**) but data is plotted on a per sample basis and for additional control datasets (serum-starved, immortalized, intermediate passage and early passage). N=3 per group.

1. Fig. 5B shows the opposite of the authors claims: in the control samples there are more transposon reads than in the KCl samples.

Thank you for pointing this out. During preparation of the manuscript the labels of Fig. 5B were switched (however, the color matching between Fig. 5A-C is correct). We apologize for this mistake, which we have now corrected.

1. "induced readthrough led to preferential expression of gene proximal transposons (i.e. those within 25 kb of genes), when compared with senescence or aging". A convincing analysis would show if there is indeed preferential proximity of induced transposons to TSEs. Since readthrough transcription decays as a function of distance from TSEs, the expression of transposons should show the same trends if indeed simply caused by readthrough. Also, these should be compared to the extent of transposon expression (not induction) in intergenic regions without any readthrough, in these conditions.

This is a very good suggestion. We now provide two new supplementary figures analyzing the distance-dependence of transposon expression.

In the first figure (Fig. S13) we show that readthrough decreases with distance (A, B) and we show that transposon counts are higher for transposons close to genes, following a similar pattern to readthrough. This is true in fibroblasts isolated from aged donors (A) and with cellular senescence (B).

**Author response image 8. sa2fig7:** Readthrough counts (rt_counts) decrease exponentially downstream of genes, both in the aging dataset (A) and in the cellular senescence dataset (B). Although noisier, the pattern for transposon counts (transp_cum_counts) is similar with higher counts closer to gene terminals, both in the aging dataset (C) and in the cellular senescence dataset (D). Readthrough counts are the cumulative counts across all genes and samples. Readthrough was determined in 10 kb bins and the values are assigned to the midpoint of the bin for easier plotting. Transposon counts are the cumulative counts across all samples for each transposon that did not overlap a neighboring gene. n=801 in (C) and n=3479 in (D).

In the second figure (Fig. S14) we show that transposons found downstream of genes with high readthrough show a more pronounced log-fold change (differential expression) than transposons downstream of genes with low readthrough (defined based on log-fold change). This is true in fibroblasts isolated from aged donors (A) and with cellular senescence (B). Furthermore, the difference between high and low readthrough region transposons is diminished for transposons that are more than 10 kb downstream of genes, as would be expected given that readthrough decreases with distance.

**Author response image 9. sa2fig8:** Transposons found downstream of genes with high readthrough (hi_RT) show a more pronounced log-fold change (transp_logfc) than transposons downstream of genes with low readthrough (low_RT). This is true in fibroblasts isolated from aged donors (A) and with cellular senescence (B). Furthermore, the difference between high and low readthrough region transposons is diminished for transposons that are more than 10 kb downstream of genes (“Transp > 10 kb”). Transposons in high readthrough regions were defined as those in the top 20% of readthrough log-fold change. Readthrough was measured between 0 and 10 kb downstream from genes. n=2124 transposons in (A) and n=6061 transposons in (B) included in the analysis.

**Reviewer #2 (Public Review):**
In this manuscript, the authors examined the role of transcription readout and intron retention in increasing transcription of transposable elements during aging in mammals. It is assumed that most transposable elements have lost the regulatory elements necessary for transcription activation. Using available RNA-seq datasets, the authors showed that an increase in intron retention and readthrough transcription during aging contributes to an increase in the number of transcripts containing transposable elements.Previously, it was assumed that the activation of transposable elements during aging is a consequence of a gradual imbalance of transcriptional repression and a decrease in the functionality of heterochromatin (de repression of transcription in heterochromatin). Therefore, this is an interesting study with important novel conclusion. However, there are many questions about bioinformatics analysis and the results obtained.Major comments:1. In Introduction the authors indicated that only small fraction of LINE-1 and SINE elements are expressed from functional promoters and most of LINE-1 are co-expressed with neighboring transcriptional units. What about other classes of mobile elements (LTR mobile element and transposons)?

We thank the reviewer for this comment. Historically, most repetitive elements, e.g. DNA elements and retrotransposon-like elements, have been considered inactive, having accrued mutations which prevent them from transposition. On the other hand, based on recent data it is indeed very possible that certain LTR elements become active with aging as suggested in several manuscripts (Liu et al. 2023, Autio et al. 2020). However, these elements are not well annotated and our final analysis (Fig. 6) relies on a well-defined distinction between active and inactive elements. (See also question 2 for further discussion.)

Finally, we would like to point out some of the difficulties with defining expression and re-activation of LTR/ERV elements based on RNAseq data that have been highlighted for the Liu manuscript and are concordant with several of our results:https://pubpeer.com/publications/364E785636ADF94732A977604E0256

Liu, Xiaoqian, et al. "Resurrection of endogenous retroviruses during aging reinforces senescence." Cell 186.2 (2023): 287-304.

Autio A, Nevalainen T, Mishra BH, Jylhä M, Flinck H, Hurme M. Effect of ageing on the transcriptomic changes associated with expression at the HERV-K (HML-2) provirus at 1q22. Immun Ageing. 2020;17(1):11.

1. Results: Why authors considered all classes of mobile elements together? It is likely that most of the LTR containing mobile elements and transposons contain active promoters that are repressed in heterochromatin or by KRAB-C2H2 proteins.

We do not consider LTR containing elements because there is uncertainty regarding their overall expression levels and their expression with aging (Nevalainen et al. 2018). Furthermore, we believe that substantial activity of LTR elements in human genomes should have been detectable through patterns of insertional mutagenesis. Yet studies generally show low to negligible levels of LTR (ERV) mutagenesis. Here, for example, at a 200-fold lower rate than for LINEs (Lee et al. 2012).

Importantly, our analysis in Fig. 6 relies on well-annotated elements like LINEs, which is why we do not include LTR or SINE elements that could be potentially expressed. However, for other analyses we did consider element families independently as can be seen in Table S1, for example.

Nevalainen, Tapio, et al. "Aging-associated patterns in the expression of human endogenous retroviruses." PLoS One 13.12 (2018): e0207407.

Lee, Eunjung, et al. "Landscape of somatic retrotransposition in human cancers." Science 337.6097 (2012): 967-971.

1. Fig. 2. A schematic model of transposon expression is not presented clearly. What is the purpose of showing three identical spliced transcripts?

This is indeed confusing. There are three spliced transcripts to schematically indicate that the majority of transcripts will be correctly spliced and that intron retention is rare (estimated at 4% of all reads in our dataset). We have clarified the figure now, please see below:

**Author response image 10. sa2fig9:** A schematic model of transposon expression. In our model, represented in this schematic, transcription (**A**) can give rise to mRNAs and pre-mRNAs that contain retained introns when co-transcriptional splicing is impaired. This is often seen during aging and senescence, and these can contain transposon sequences (**B**). In addition, transcription can give rise to mRNAs and pre-mRNAs that contain transposon sequences towards the 3’-end of the mRNA when co-transcriptional termination at the polyadenylation signal (PAS) is impaired (**C, D**) as seen with aging and senescence. Some of these RNAs may be successfully polyadenylated (as depicted here) whereas others will be subject to nonsense mediated decay. Image created with Biorender.

1. The study analyzed the levels of RNA from cell cultures of human fibroblasts of different ages. The annotation to the dataset indicated that the cells were cultured and maintained. The cells were cultured in high-glucose (4.5mg/ml) DMEM (Gibco) supplemented with 15% (vol/vol) fetal bovine serum (Gibco), 1X glutamax (Gibco), 1X non-essential amino acids (Gibco) and 1% (vol/vol) penicillin-streptomycin (Gibco). How correct that gene expression levels in cell cultures are the same as in body cells? In cell cultures, transcription is optimized for efficient division and is very different from that of cells in the body. In order to correlate a result on cells with an organism, there must be rigorous evidence that the transcriptomes match.

We agree and have updated the discussion to reflect this shortcoming. While we do not have human tissue data, we would like to draw the reviewer’s attention to Fig. S3 where we presented some liver data for mice. We now provide an additional supplementary figure (in a style similar to Fig. S2) showing how readthrough, transposon expression and intron retention changes in 26 vs 5-month-old mice (Fig. S4). Indeed, intron, readthrough and transposons increase with age in mice, although this is more pronounced for transposons and readthrough.

**Author response image 11. sa2fig10:** Intron, readthrough and transposon elements are elevated in the liver of aging mice (26 vs 5-month-old, n=6 per group). Readthrough and transposon expression is especially elevated even when compered to genic transcripts. The percentage of upregulated transcripts is indicated above each violin plot and the median log10-fold change for genic transcripts is indicated with a dashed red line.

Finally, just to elaborate, we used the aging fibroblast dataset by Fleischer et al. for three reasons:

1. Yes, aging fibroblasts could be a model of human aging, with important caveats as you correctly point out,

2. it is one of the largest such datasets allowing us to draw conclusions with higher statistical confidence and do things such as partial correlations

3. it has been analyzed using similar techniques before (LaRocca, Cavalier and Wahl 2020) and this dataset is often used to make strong statements about transposons and aging such as transposon expression in this dataset being “consistent with growing evidence that [repetitive element] transcripts contribute directly to aging and disease”. Our goal was to put these statements into perspective and to provide a more nuanced interpretation.

LaRocca, Thomas J., Alyssa N. Cavalier, and Devin Wahl. "Repetitive elements as a transcriptomic marker of aging: evidence in multiple datasets and models." Aging Cell 19.7 (2020): e13167.

1. The results obtained for isolated cultures of fibroblasts are transferred to the whole organism, which has not been verified. The conclusions should be more accurate.

We agree and have updated the discussion accordingly.

1. The full pipeline with all the configuration files IS NOT available on github (pabisk/aging_transposons).

Thank you for pointing this out, we have now uploaded the full pipeline and configuration files.

1. Analysis of transcripts passing through repeating regions is a complex matter. There is always a high probability of incorrect mapping of multi-reads to the genome. Things worsen if unpaired short reads are used, as in the study (L=51). Therefore, the authors used the Expectation maximization algorithm to quantify transposon reads. Such an option is possible. But it is necessary to indicate how statistically reliable the calculated levels are. It would be nice to make a similar comparison of TE levels using only unique reads. The density of reads would drop, but in this case it would be possible to avoid the artifacts of the EM algorithm.

We thank the reviewer for this suggestion. We show here that mapping only unique alignments (outFilterMultimapNmax=1 in STAR) leads to similar results.

For the aging fibroblast dataset:

**Author response image 12. sa2fig11:** 

For the induced senescence dataset:

**Author response image 13. sa2fig12:**